# Serum Levels of TNF-α Are Increased in Patients with Rotator Cuff Tear and Sleep Disturbance

**DOI:** 10.3390/diagnostics11122215

**Published:** 2021-11-27

**Authors:** Chul-Hyun Cho, Du-Hwan Kim, Eun-Hee Baek, Du-Han Kim

**Affiliations:** 1Department of Orthopedic Surgery, Keimyung University Dongsan Hospital, Keimyung University School of Medicine, Daegu 42601, Korea; whitelioneh@hanmail.net (E.-H.B.); osmdkdh@gmail.com (D.-H.K.); 2Department of Physical Medicine and Rehabilitation, College of Medicine, Chung-Ang University, Seoul 06973, Korea; ri-pheonix@hanmail.net

**Keywords:** rotator cuff, sleep, cytokine, tumor necrosis factor-α

## Abstract

The purpose of this study was to determine serum levels of sleep-related cytokines in patients with rotator cuff tear (RCT) who were experiencing pain-related sleep disturbance. Peripheral blood samples before surgery were collected from 63 study participants and divided into three groups: RCT with sleep disturbance group; SD group (*n* = 21), RCT with normal sleep group; NS group (*n* = 21), and patients with chronic shoulder instability; control group (*n* = 21). Serum concentration levels of interleukin-1α (IL-1α), IL-1β, IL-2, IL-6, IL-8, IL-10, and tumor necrosis factor-α (TNF-α) were measured via ELISA. The associations between serum levels of sleep-related cytokines and clinical scores and the Pittsburgh Sleep Quality Index (PSQI) were analyzed. Serum concentration levels of TNF-α were significantly higher in the SD group compared with those of the NS and control groups (*p* = < 0.001 and 0.05). Serum levels of IL-8 and IL-10 were significantly higher in the SD group compared with those of control group (*p* = 0.01 and = 0.05), but did not differ significantly from that of the NS group. There were no associations between serum levels of sleep-related cytokines and all clinical scores. The current findings suggest that TNF-α may be associated with sleep disturbance in patients with RCT.

## 1. Introduction

Rotator cuff tear (RCT) is the most common cause of shoulder pain and functional disability leading to a considerable socioeconomic burden [1,2]. Although intrinsic and extrinsic processes have been suggested as the underlying cause of RCT, the exact etiology and pathophysiology leading to the disease remain controversial [3]. It has been widely reported that bursitis of the subacromial space (SAS) and synovitis of the glenohumeral joint (GHJ) both play a role in the development of shoulder pain in patients with RCT and that the severity correlates with the pain intensity [4,5,6,7]. Several studies have reported that clinical symptoms and disease progression are involved in overexpression of proinflammatory and pain-related cytokines in the shoulder joint, including subacromial bursa, joint capsule, and joint fluid [3,4,5,6,7,8]. Additionally, previous findings indicate that inflammatory mediators such as interleukin-1α (IL-1α), IL-1β, IL-6, IL-8, tumor necrosis factor-α (TNF-α), cyclooxygenase (COX)-1, and COX-2 play an important role in inflammation and pain due to RCT [3,4,5,6,7,8].

Night pain is a common symptom in patients with shoulder problems including RCT, frozen shoulder, and osteoarthritis [9,10]. Cho et al. [9] found that 81.5% of patients with chronic shoulder pain reported sleep disturbance due to night pain and the prevalence was higher than patients with low back pain, fibromyalgia, or rheumatoid arthritis. However, the etiology and pathophysiology for the occurrence of night pain and sleep disturbance in these patients remain largely unknown.

Sleep regulation is complex and is associated with changes in the expression of numerous molecules that are directly involved in sleep [11,12,13,14,15]. Sleep disturbance has been associated with changes in cytokines, which are molecules used for intercellular communication by the immune system. Emerging evidence suggests that sleep disturbance can sensitize or dysregulate inflammatory pathways [16]. Furthermore, previous studies have demonstrated that sleep disturbance in various diseases is associated with altered serum levels of sleep-related cytokines including IL-1, IL-2, IL-6, IL-8, IL-10, and TNF-α [16,17,18,19,20,21,22,23]. These cytokines are also involved in inflammatory, anti-inflammatory, and pain pathways.

Sleep problems with night pain in patients with shoulder disease can create a vicious cycle of persistent pain perception, sleep disturbance, daytime dysfunction, behavioral or emotional changes, depression, and anxiety [9,10]. Sleep disturbance is a clinically significant target as it can be effectively treated. Despite the clinical importance, the biological underlying mechanism for sleep disturbance in patients with RCT is poorly understood. No prior studies have investigated serum levels of sleep-related cytokines to elucidate the pathophysiologic process of sleep disturbance in patients with RCT.

The primary purpose of this study was to determine serum concentration levels of sleep-related cytokines in patients with RCT and sleep disturbance. The secondary purpose was to investigate the correlations between serum levels of sleep-related cytokines and clinical scores. This study was conducted to test the hypothesis that serum levels of sleep-related cytokines are increased in patients with RCT who are experiencing pain-related sleep disturbance.

## 2. Materials and Methods

### 2.1. Subjects

This research has been approved by the IRB of the authors’ affiliated institutions and informed consent was obtained from all patients. Study subjects included 21 RCT patients with sleep disturbance (sleep disturbance group; SD group) and 21 RCT patients without sleep disturbance (normal sleep group; comparative group; NS group) who were undergoing arthroscopic surgery for RCT and 21 patients (control group) who were undergoing arthroscopic labral repair for chronic anterior shoulder instability. The control group included patients that had recurrent shoulder instability without pain. Exclusion criteria included: (1) traumatic RCT, (2) a history of insomnia, (3) a history of psychiatric disorders, and (4) the presence of antipsychotics or sedative drug medication.

### 2.2. Assessments and Instruments

Before surgery, 42 participants with RCT were evaluated using the Visual Analogue Scale (VAS) pain score, the University of California at Los Angeles (UCLA) score, and the Pittsburgh Sleep Quality Index (PSQI). This PSQI is a 19-item, self-report, questionnaire-based index that obtains information from the patient regarding sleep habits during the preceding month to measure subjective sleep quality [24]. This questionnaire scores subjective sleep quality, sleep latency, sleep duration, habitual sleep efficiency, sleep disturbances, use of sleep medication, and daytime dysfunction. Sleep disturbance was defined as a total score of ≥5.

### 2.3. Enzyme-Linked Immunosorbent Assay

Peripheral blood samples were collected in the operating room prior to surgery, between 8 a.m. and 12 p.m. Collected blood samples were centrifuged to isolate the serum, which was harvested and stored at −20 °C for batch analysis. The sleep-related serum markers selected for this study included IL-1α, IL-1β, IL-2, IL-6, IL-8, IL-10, and TNF-α based on results from previous studies [16,17,18,19,20,21,22,23].

Serum concentration levels of sleep-related cytokines including IL-1α, IL-1β, IL-2, IL-6, IL-8, IL-10, and TNF-α were detected using a Magnetic Luminex Screening Assay (Luminex, Austin, TX, USA) according to the manufacturer’s protocol. The microparticle cocktail (50 μL) was added into a 96-well plate, followed by 50 μL of standard and diluted samples. The microplates were incubated for 2 h at RT with gentle shaking. A volume of 50 μL diluted biotin antibody cocktail and diluted streptavidin–phycoerythrin were added and incubated at RT with shaking. The microparticles were resuspended with 100 μL wash buffer. Following incubation for 2 min, the microplates were read using the Luminex. Serum markers were assayed in duplicate and all values were expressed as the mean of the two measurements. The data for serum concentration levels of IL-1α, IL-1β, and IL-2 were excluded in this study because any value was out of the standard curve.

### 2.4. Statistical Methods

Statistical analysis of data was performed using SPSS (version 20.0; IBM, Armonk, NY, USA). Data were analyzed using the Mann–Whitney-U test and chi-squared test. To determine the correlations between serum levels of sleep-related cytokines and clinical scores, the Kendal tau B correlation analysis was used. The results are expressed as mean values ± standard deviation (SD) and were considered significantly different when the two-tailed *p* value was < 0.05.

## 3. Results

The mean age of patients was 56.9 ± 8.6 years in the SD group, 55.7 ± 7.3 years in the NS group, and 30.1 ± 13.1 years in the control group. There were 12 women and 9 men in the SD group, 11 women and 10 men in the NS group, and 3 women and 18 men in the control group. No statistically significant differences were found between the SD group and the NS group regarding age, sex, dominant arm, diabetes mellitus, duration of symptoms, preoperative stiffness, tear size, preoperative VAS pain score, or preoperative UCLA score (*p* > 0.05). Significant differences were found in the mean preoperative PSQI score, which was 7.4 ± 2.9 in the SD group and 3.0 ± 1.1 in the NS group (*p* < 0.001) (Table 1).

### 3.1. IL-6

Serum concentration levels of IL-6 were 0.31 ± 0.31 pg/mL in the SD group, 1.26 ± 0.64 pg/mL in the NS group, and 2.75 ± 1.45 pg/mL in the control group. Serum level of IL-6 was significantly lower in the SD group compared with those of the NS and control groups (*p* = < 0.001 and 0.01) (Figure 1).

### 3.2. IL-8

Serum concentration levels of IL-8 were 22.30 ± 6.04 pg/mL in the SD group, 17.62 ± 2.88 pg/mL in the NS group, and 13.26 ± 2.26 pg/mL in the control group. Serum levels of IL-8 were significantly higher in the SD group compared with those of the control group (*p* = 0.01), but did not differ significantly from that of the NS group (*p* > 0.05) (Figure 2).

Serum concentration levels of IL-10 were 2.94 ± 0.42 pg/mL in the SD group, 2.93 ± 0.28 pg/mL in the NS group, and 2.65 ± 0.38 pg/mL in the control group. Serum levels of IL-10 were significantly higher in the SD group compared with those of the control group (*p* = 0.05), but did not differ significantly from that of the NS group (*p* > 0.05) (Figure 3).

### 3.3. TNF-α

Serum concentration levels of TNF-α were 4.24 ± 0.56 pg/mL in the SD group, 3.08 ± 0.58 pg/mL in the NS group, and 3.67 ± 0.58 pg/mL in the control group. Serum levels of TNF-α were significantly higher in the SD group compared with those of the NS and control groups (*p* = < 0.001 and 0.05) (Figure 4).

### 3.4. Correlations between Serum Levels of Sleep-Related Cytokines and Clinical Scores

Correlation analyses revealed that there were no associations between serum levels of sleep-related cytokines and all clinical scores including VAS pain, UCLA, and PSQI scores (*p* > 0.05) (Table 2).

## 4. Discussion

This present study was conducted to test the hypothesis that serum levels of sleep-related cytokines are increased in patients with RCT who are experiencing pain-related sleep disturbance. To the best of our knowledge, this is the first study to investigate levels of sleep-related cytokines in patients with RCT. Our study demonstrated that serum concentration levels of TNF-α were significantly higher in the SD group compared with those of the NS and control groups. There were no associations between serum levels of sleep-related cytokines and clinical scores including VAS pain, UCLA, and PSQI scores. Our findings suggest that TNF-α may be associated with sleep disturbance in patients with RCT.

Numerous studies have demonstrated the association between poor sleep, inflammatory cytokines, and pain [11,12,15,16,17,22]. Clinton et al. [11] reported that sleep deprivation and altered cytokine levels are associated with increased sensitivity to pain with sleepiness and rebound sleep. Heffner et al. [16] also reported that inflammatory and pain pathways can be the underlying mechanism responsible for the association between chronic pain and sleep disturbance. In patients with chronic pain such as fibromyalgia, chronic low back pain, osteoarthritis, or chronic fatigue syndrome, sleep disturbance was found to have significant associations with altered levels of cytokines [16]. Previous studies have demonstrated that sleep disturbance in various diseases is associated with altered circulating levels of sleep-related cytokines including IL-1, IL-2, IL-6, IL-8, IL-10, and TNF-α [16,17,18,19,20,21,22,23]. These cytokines are also involved in inflammatory, anti-inflammatory, and pain pathways. Especially, IL-1α, IL-1β, IL-6, IL-8, and TNF-α have been found to play an important role in inflammation and pain due to RCT [3,4,5,6,7,8]. In comprehensive review of results from previous studies related sleep disturbance in various diseases, the present study included IL-1α, IL-1β, IL-2, IL-6, IL-8, IL-10, and TNF-α as sleep-related serum markers.

IL-1, IL-6, and TNF-α are pleiotropic, serving both physiologic and pathological functions including modulation of inflammation, sleep, and mood [11]. Several studies reported that sleep deprivation is associated with increased levels of IL-6 and TNF-α [12,20]. Additionally, poor sleep quality and increased frequency of sleep disturbance have been associated with elevated levels of IL-2 and IL-6 [12,20]. Lee et al. [20] found that elevated levels of proinflammatory cytokines such as IL-6 and TNF-α were associated with poorer sleep quality in patients with schizophrenia. Cytokines including IL-1 and TNF-α partake in non-rapid eye movement sleep (NREMS) regulation under physiological and inflammatory conditions [11,13].

Rockstrom et al. [15] found that TNF-α is involved in sleep regulation, acting within an extensive, tightly orchestrated biochemical network impacting sleep in health and disease. Clinton et al. [11] also described that cytokines such as IL-1 and TNF-α play a role in sleep regulation as they are currently the best-characterized sleep regulatory substances. Circulating levels of TNF-α are altered in many pathologies that exhibit sleep disturbances, including chronic inflammation, rheumatoid arthritis, insomnia, and chronic fatigue syndrome [14]. Rockstrom et al. [15] reported that high circulating levels of TNF-α are associated with sleep problems and normalization of high TNF-α level improves sleep. Many symptoms induced by sleep loss, e.g., sleepiness, fatigue, poor cognition, enhanced pain sensitivity, can be elicited with an injection of exogenous IL-1 or TNF-α [11,13,14]. Clinically available inhibitors of IL-1 and TNF-α reduce the sleepiness and fatigue associated with rheumatoid arthritis and sleep apnea [11]. The present study demonstrated that serum concentration levels of TNF-α were significantly higher in the SD group compared with those of the NS and control groups. Our findings support that TNF-α may play a significant role in pathophysiology of sleep disturbance as well as pain mechanisms in patients with RCT, and may be a possible therapeutic target to improve sleep disturbance.

Heffner et al. [17] found that poorer sleep quality is associated with higher circulating levels of IL-6 in patients with chronic low back pain. Additionally, Ji et al. [18] found that IL-6 levels are positively correlated with cluster symptoms of pain, fatigue, depression, and sleep disturbance in patients with cancer. Wang et al. [23] found elevated IL-6 levels are associated with sleep disturbance in patients with major depressive disorder. However, the present study found that serum levels of IL-6 were significantly lower in the SD group compared with those of the NS and control groups. These are apparently contradictory results compared to those of previous studies. Proinflammatory and anti-inflammatory cytokines influence physiological sleep and sleep responses to pathological insult [15]. IL-8 is a multifunctional chemokine that has a strong influence on the activation, regulation, and chemotactic effect of neutrophils and is significantly increased in patients with sleep disorders and various inflammatory conditions [21]. IL-10, also known as human cytokine synthesis inhibitory factor, is an anti-inflammatory cytokine. Kim et al. [19] found that circulating IL-10 levels in those with idiopathic rapid eye movement sleep (REMS) disorder were significantly upregulated compared to those without idiopathic REMS disorder. Park et al. [22] found that increased IL-10 levels were related to low sleep quality in patients with temporomandibular disorder, supporting the finding that anti-inflammatory cytokines have a sleep-disrupting effect. Although the present study revealed serum levels of IL-8 and IL-10 were significantly higher in the SD group compared with those of the control group, differences were not significant when compared to the NS group. In light of these findings, further well-designed studies are needed to elucidate the connections between sleep disturbance and sleep-related cytokines such as IL-6, IL-8, and IL-10 in patients with RCT.

Recently, Ha et al. [10] reported that melatonin may play a role as a mediator of night pain in patients with RCT or frozen shoulder. They found that melatonin increases the production of proinflammatory and pain-related cytokines such as IL-1α, IL-1β, IL-2, IL-6, and TNF-α [10]. Circadian changes in melatonin levels may be an important contributor to the pathophysiology and treatment of patients with RCT. Further studies are needed to provide a better understanding of the role of melatonin and cytokines on symptoms including pain, inflammation, and sleep disturbance.

This study has several limitations. First, the control group was not matched for age and sex when compared to the SD group. However, the NS group was provided as an additional comparison group. Second, the data for serum concentration levels of IL-1α, IL-1β, and IL-2 were excluded in this study because any value was out of the standard curve. Third, this is a cross-sectional study, and serial changes of sleep-related cytokines were not compared with symptoms after the operative procedure. Future prospective longitudinal studies with a wider spectrum of sleep-related mediators are needed to clarify these issues.

## 5. Conclusions

Our study demonstrated that serum concentration levels of TNF-α are increased in patients with RCT who are experiencing pain-related sleep disturbance. The current findings suggest that TNF-α may be associated with sleep disturbance in patients with RCT.

## Figures and Tables

**Figure 1 diagnostics-11-02215-f001:**
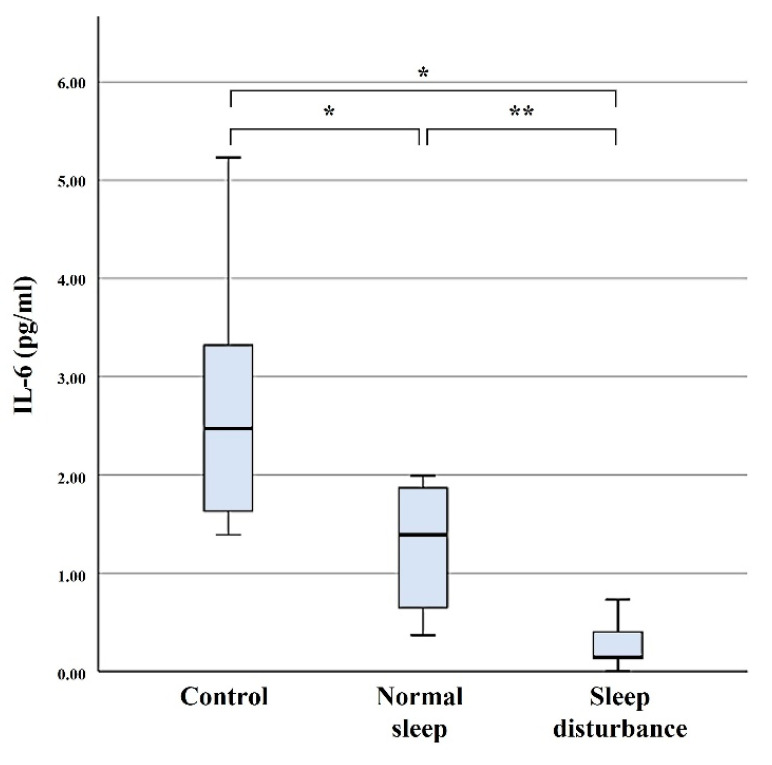
Serum concentration levels of IL-6. The error bars indicate the 95% confidence interval. * *p* < 0.05; ** *p* < 0.001.

**Figure 2 diagnostics-11-02215-f002:**
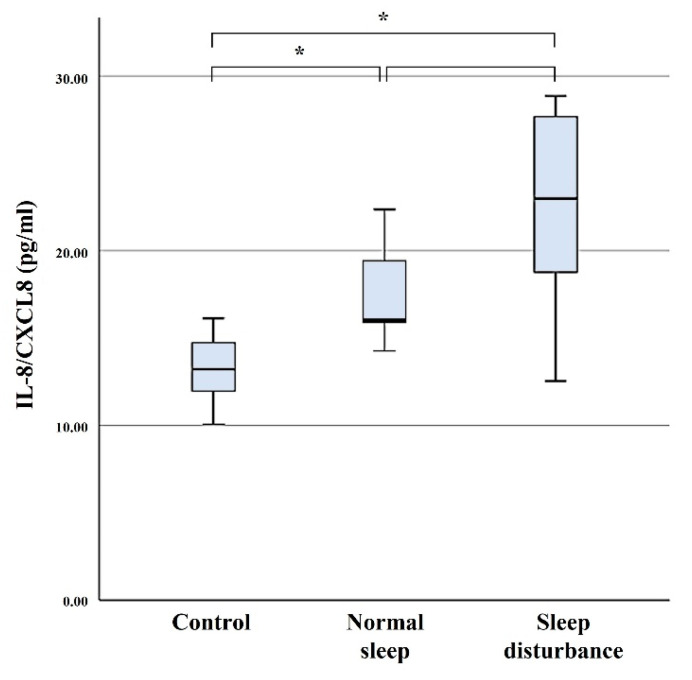
Serum concentration levels of IL-8. The error bars indicate the 95% confidence interval. * *p* < 0.05.

**Figure 3 diagnostics-11-02215-f003:**
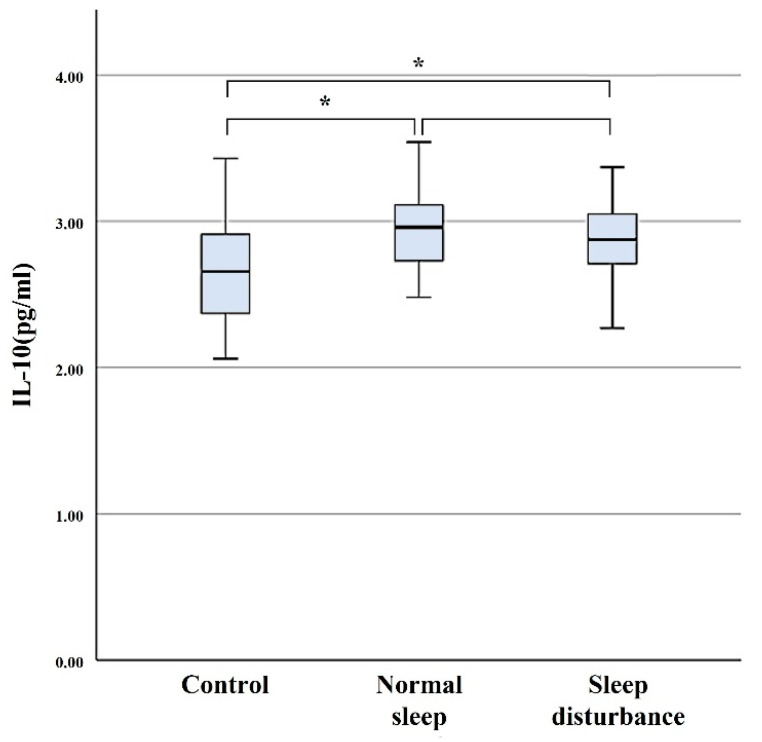
Serum concentration levels of IL-10. The error bars indicate the 95% confidence interval. * *p* < 0.05.

**Figure 4 diagnostics-11-02215-f004:**
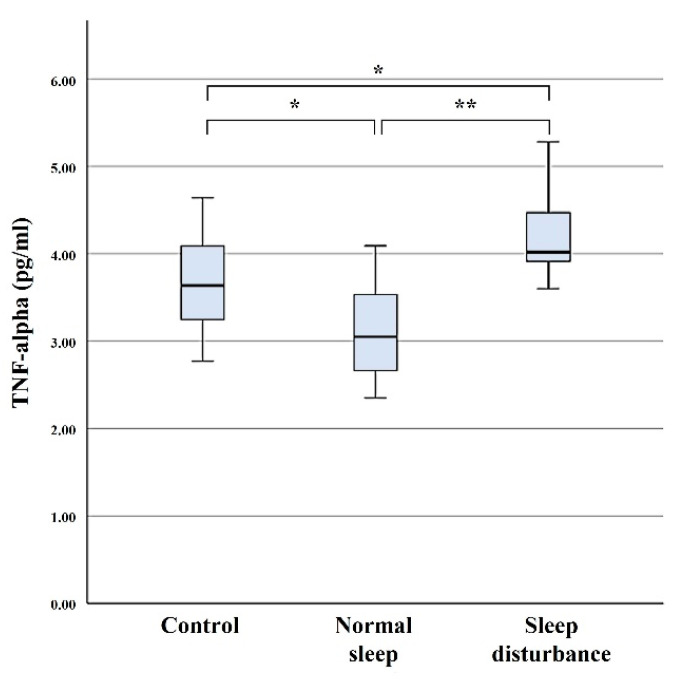
Serum concentration levels of TNF-α. The error bars indicate the 95% confidence interval. * *p* < 0.05; ** *p* < 0.001.

**Table 1 diagnostics-11-02215-t001:** Demographic and clinical data.

	Sleep Disturbance Group (*n* = 21)	Normal Sleep Group (*n* = 21)	*p* Value
Age, year	56.9 ± 8.6	55.7 ± 7.3	0.521
Sex, male/female, *n*	9/12	10/11	0.757
Dominant arm, yes/no, *n*	14/7	13/8	0.747
Diabetes mellitus, yes/no, *n*	4/17	2/19	0.663
Duration of symptoms, months	29.09 ± 40.1	28.5 ± 41.3	0.533
Tear size (partial/small/medium/large/massive)	12/0/2/3/4	8/4/4/2/3	0.214
Preoperative stiffness, yes/no, *n*	8/13	10/11	0.284
Preoperative VAS pain score	6.9 ± 1.2	6.1 ± 1.8	0.071
Preoperative UCLA score	13.5 ± 4.3	13.9 ± 5.1	0.640
Preoperative PSQI score	7.4 ± 2.9	3.0 ± 1.1	<0.001 *

VAS, visual analog scale; UCLA, University of California, Los Angeles; PSQI; Pittsburgh Sleep Quality Index. * Statistically significant, *p* < 0.05.

**Table 2 diagnostics-11-02215-t002:** Correlation analyses between serum levels of sleep-related cytokines and clinical scores.

		VAS Pain Score	UCLA Score	PSQI Score
IL-6	Coefficient	0.177	0.003	0.079
	*p* value	0.482	0.992	0.755
IL-8	Coefficient	−0.794	0.609	0.580
	*p* value	0.059	0.200	0.228
IL-10	Coefficient	−0.349	0.392	−0.050
	*p* value	0.131	0.087	0.835
TNF-α	Coefficient	0.434	−0.418	0.330
	*p* value	0.330	0.350	0.469

VAS, visual analog scale; UCLA, University of California, Los Angeles; PSQI, Pittsburgh Sleep Quality Index.

## Data Availability

Not applicable.

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
