# Peer review of "Serum Levels of TNF-α Are Increased in Patients with Rotator Cuff Tear and Sleep Disturbance"

_diagnostics, 2021, doi:10.3390/diagnostics11122215_

Round 1
Reviewer 1 Report
- The control group was selected from patients who underwent shoulder arthroscopy due to instability. There is lack of data about the time from trauma to treatment, the exact treatment of them, acute or chronic instability, additional lesions, time from trauma to arthroscopy. This is not a control group because is can be biased by possible problems with SSC tendon due to trauma during the anterior dislocation or Hill Sachs lesion or SLAP e.g. Authors should reconsider this.
- The is no info about the cuff lesions and their characteristic, single tendon or massive lesions, acute or chronic, biceps involve or not. Traumatic cuff lesions or degenerative? This data must be improved.
3. What about the concentration of Il-1 and 2 in results, they are mentioned in methods section.
4. The conclusions do not support the results.
Author Response
Dear. Reviewer 1
Thank you for your sincere response.
According to your comments, we did our best to revise our paper. We believe that this process makes our study more valuable. Our answers to your comments are below.
<Response to Reviewer 1 Comments>
- The control group was selected from patients who underwent shoulder arthroscopy due to instability. There is lack of data about the time from trauma to treatment, the exact treatment of them, acute or chronic instability, additional lesions, time from trauma to arthroscopy. This is not a control group because is can be biased by possible problems with SSC tendon due to trauma during the anterior dislocation or Hill Sachs lesion or SLAP e.g. Authors should reconsider this. --> Thanks for your comment. As control group, we included the patients with chronic anterior shoulder instability. These patients had symptoms of recurrent shoulder instability without pain. Most previous basic studies for rotator cuff disease have used the patients with shoulder instability as control group. According to your comment, we revised and added the sentences “21 patients (control group) who were undergoing arthroscopic labral repair for chronic anterior shoulder instability. Control group included the patients had recurrent shoulder instability without pain.”. Please see Page 2, Line 76-78.
- The is no info about the cuff lesions and their characteristic, single tendon or massive lesions, acute or chronic, biceps involve or not. Traumatic cuff lesions or degenerative? This data must be improved. --> Thanks for your comment. According to your comment, we added them in method section and Table I. Please see Page 2, Line 79 and Page 3, Table 1.
- What about the concentration of Il-1 and 2 in results, they are mentioned in methods section. --> Thanks for your comment. According to your comment, we added this sentence. “The data for serum concentration levels of IL-1α, IL-1β, and IL-2 were excluded in this study because any value was out of the standard curve.” Please see page 3, Line 105-106.
- The conclusions do not support the results. --> Thanks for your comment. We agree with your opinion. According to your comment, we revised the conclusions based on our results. “Our study demonstrated that serum concentration levels of TNF-α are increased in patients with RCT who are experiencing pain-related sleep disturbance. The current findings suggest that TNF-α may be associated with sleep disturbance in patients with RCT.” Please see the Page 9, line 266-268.

Reviewer 2 Report
Thank you for the opportunity to review the manuscript entitled "TNF-α plays a role in sleep disturbance in patients with rotator cuff tear". This is a cross-sectional study in which the serum levels of cytokines are compared in three groups of patients, two of them with rotated cuff tear, with or without sleep disorder, and a third group with shoulder instability. The topic of the work is very interesting, although the manuscript presents some methodological problems.
The main problem of the work is a certain lack of coherence between the title and the objectives and hypotheses, which are expressed differently in the abstract, the introduction and the discussion. Specific concerns are listed below.
Abstract
On page 1, lines 11 and 12, it should be noted that the patients in the first two groups were diagnosed with a rotator cuff tear, as well as their pre or post-surgical situation.
Abbreviations for “IL” should be previously defined as “interleukin”. The abbreviation TNF should also be described as Tumor Necrosis Factor.
Introduction
The authors state that the main objective of the study is to determine the serum concentration levels of sleep-related cytokines in patients with RCT, but the hypothesis is more specific as it postulates that cytokine concentrations are elevated in patients with RCT who present sleep disorder. This imprecision creates confusion for the reader who is confronted with the manuscript for the first time. Greater consistency would be required.
Materials and Methods
On page 3 (statistical methods) I don't understand why the one-way ANOVA was not used. The Mann-Whitney U is used for two-sample difference, but there are three groups in the study. And the correlation analysis with Kendal's Tau B is questionable. Other coefficients such as Pearson's r or Spearman's rho would be more convenient, depending on whether or not the parametric assumptions are fulfilled. All of this should be adequately detailed.
Results
The differences in the demographic characteristics of the control group with the two groups of patients with RCT are so important that any conclusion is risky and could be due to these differences. However, the authors already highlight this issue in the limitations of the study.
In the figures, the error bars should be represented by the 95% confidence interval, which is the standard criterion. It is striking that there is a certain overlap of the error bars, which is not usually compatible with the existence of statistically significant differences.
In general, it seems that the most obvious differences do not appear in TNF but also in IL-6 and in Il-8, although the importance is not given due to this result. Can the authors explain why?
It is very strange that very strong correlations close to 0.8 are not statistically significant. Perhaps using other correlation coefficients, the result could be more interpretable.
Discussion
At the end of the first paragraph, a causal relationship between TNF levels and sleep disturbance is practically interpreted. Given the cross-sectional nature of the study, the authors should be more cautious in interpreting the results.
Conclusion
The conclusions seem too ambitious for the type of study, given its cross-sectional methodology.
Author Response
Dear. Reviewer 2
Thank you for your sincere response.
According to your comments, we did our best to revise our paper. We believe that this process makes our study more valuable. Our answers to your comments are below.
<Response to Reviewer 2 Comments>
Thank you for the opportunity to review the manuscript entitled "TNF-α plays a role in sleep disturbance in patients with rotator cuff tear". This is a cross-sectional study in which the serum levels of cytokines are compared in three groups of patients, two of them with rotated cuff tear, with or without sleep disorder, and a third group with shoulder instability. The topic of the work is very interesting, although the manuscript presents some methodological problems.
The main problem of the work is a certain lack of coherence between the title and the objectives and hypotheses, which are expressed differently in the abstract, the introduction and the discussion. Specific concerns are listed below. --> Thanks for your comment. According to your comment, we revised the title, the objectives and hypotheses, and the conclusions for coherence.
<Abstract>
On page 1, lines 11 and 12, it should be noted that the patients in the first two groups were diagnosed with a rotator cuff tear, as well as their pre or post-surgical situation. --> Thanks for your comment. According to your comment, we revised sentence. “Peripheral blood samples before surgery were collected from 63 study participants were divided into three groups: RCT with sleep disturbance group; SD group (n = 21), RCT with normal sleep group; NS group (n = 21), and patients with chronic shoulder instability; control group (n = 21).” Please see Page 1, Line 12-15.
Abbreviations for “IL” should be previously defined as “interleukin”. The abbreviation TNF should also be described as Tumor Necrosis Factor. --> Thanks for your comment. According to your comment, we revised them. Please see Page 1, Line 16-17.
<Introduction>
The authors state that the main objective of the study is to determine the serum concentration levels of sleep-related cytokines in patients with RCT, but the hypothesis is more specific as it postulates that cytokine concentrations are elevated in patients with RCT who present sleep disorder. This imprecision creates confusion for the reader who is confronted with the manuscript for the first time. Greater consistency would be required. --> Thanks for your comment. According to your comment, we revised them. “The primary purpose of this study was to determine serum concentration levels of sleep-related cytokines in patients with RCT and sleep disturbance. The secondary purpose was to investigate the correlations between serum levels of sleep-related cytokines and clinical scores. This study was conducted to test the hypothesis that serum levels of sleep-related cytokines are increased in patients with RCT who are experiencing pain-related sleep disturbance.” Please see Page 2, Line64-69.
<Materials and Methods>
On page 3 (statistical methods) I don't understand why the one-way ANOVA was not used. The Mann-Whitney U is used for two-sample difference, but there are three groups in the study. And the correlation analysis with Kendal's Tau B is questionable. Other coefficients such as Pearson's r or Spearman's rho would be more convenient, depending on whether or not the parametric assumptions are fulfilled. All of this should be adequately detailed. --> Thanks for your comment. Study subjects had three groups involving 21 RCT patients with sleep disturbance (sleep disturbance group; SD group), 21 RCT patients without sleep disturbance (normal sleep group; comparative group; NS group), and 21 patients (control group) who were undergoing arthroscopic labral repair for chronic anterior shoulder instability. 21 RCT patients without sleep disturbance was comparative group. So, we analyzed the-sample difference (SD group vs NS group, SD group vs control group) using Mann-Whitney U. Correlation analysis using Kendal’s Tau B was used in our study because number of samples was lesser than 30 (nonparametric test).
<Results>
The differences in the demographic characteristics of the control group with the two groups of patients with RCT are so important that any conclusion is risky and could be due to these differences. However, the authors already highlight this issue in the limitations of the study. --> Thanks for your comment. We agree with your opinion. We already mentioned this issue. “First, the control group was not matched for age and sex when compared to the SD group. However, the NS group was provided as an additional comparison group.” Please see Page 9, Line 258-260.
In the figures, the error bars should be represented by the 95% confidence interval, which is the standard criterion. It is striking that there is a certain overlap of the error bars, which is not usually compatible with the existence of statistically significant differences. --> Thanks for your comment. We agree with your opinion. According to your comment, we remake tables with the error bars by 95% confidence interval. Please see Table 1-4.
In general, it seems that the most obvious differences do not appear in TNF but also in IL-6 and in Il-8, although the importance is not given due to this result. Can the authors explain why? --> Thanks for your comment. We agree with your opinion. According to your comment, we remake tables with the error bars by 95% confidence interval. We also provided statistically significant differences. Please see Table 1,2,4.
It is very strange that very strong correlations close to 0.8 are not statistically significant. Perhaps using other correlation coefficients, the result could be more interpretable. --> Thanks for your comment. In correlation analysis of Table 2, coefficient between IL8 and VAS pain score was -0.794 and p-value was 0.059. However, the data was considered significantly different when the two-tailed p value was < 0.05. This result may originated by small number of samples.
<Discussion>
At the end of the first paragraph, a causal relationship between TNF levels and sleep disturbance is practically interpreted. Given the cross-sectional nature of the study, the authors should be more cautious in interpreting the results. --> Thanks for your comment. We agree with your opinion. According to your comment, we revised the sentence based on our results. “Our findings suggest that TNF-α may be associated with sleep disturbance in patients with RCT.”
<Conclusion>
The conclusions seem too ambitious for the type of study, given its cross-sectional methodology. --> Thanks for your comment. We agree with your opinion. According to your comment, we revised the conclusions based on our results. “Our study demonstrated that serum concentration levels of TNF-α are increased in patients with RCT who are experiencing pain-related sleep disturbance. The current findings suggest that TNF-α may be associated with sleep disturbance in patients with RCT.” Please see the Page 9, line 266-268.

Round 2
Reviewer 1 Report
The manuscript was modified according to my guides, I have no further concerns.
Author Response
Thank you for your sincere review.
<Response to Reviewer 1 Comments>
The manuscript was modified according to my guides, I have no further concerns
--> Thank you very much.

Reviewer 2 Report
The authors have resolved most of my doubts and concerns about the manuscript. I only have some questions and suggestions.
1. I understand that the old graphics that appear in the new version of the manuscript are crossed out and will be replaced by the new graphics.
2. Also, I assume that the error bars no longer represent the standard deviation but rather the 95% confidence interval. If this is so, please modify the figure captions including this clarification.
Author Response
Thank you for your sincere review.
According to your comments, we did our best to revise our paper. We believe that this process makes our study more valuable. Our answers to your comments are below.
<Response to Reviewer 2 Comments>
The authors have resolved most of my doubts and concerns about the manuscript. I only have some questions and suggestions.
- I understand that the old graphics that appear in the new version of the manuscript are crossed out and will be replaced by the new graphics.
--> Thanks for your comment. Previously, we crossed out old graphics through track change in revised manuscript.
- Also, I assume that the error bars no longer represent the standard deviation but rather the 95% confidence interval. If this is so, please modify the figure captions including this clarification.
--> Thanks for your comment. According to your comment, we changed to “The error bars indicate the 95% confidence intervals” for Figure 1- 4 captions.
